# Perfect Reconciliation in Quantum Key Distribution with Order-Two Frames

**Luis Adrián Lizama-Pérez** [1],* and **José Mauricio López-Romero** [2]

1 Dirección de Investigación, Innovación y Posgrado, Universidad Politécnica de Pachuca, Ex-Hacienda de Santa Bárbara, Zempoala 43830, Mexico
2 Cinvestav Querétaro, Libramiento Norponiente 2000, Real de Juriquilla, Santiago de Querétaro 76230, Mexico; jm.lopez@cinvestav.mx
* Correspondence: luislizama@upp.edu.mx

**Abstract:** We present an error reconciliation method for Quantum Key Distribution (QKD) that corrects 100% of errors generated in regular binary frames transmitted over a noisy quantum channel regardless of the quantum channel error rate. In a previous investigation, we introduced a novel distillation QKD algorithm whose secret key rate descends linearly with respect to the channel error rate. Now, as the main achievement of this work, we demonstrate an improved algorithm capable of retaining almost all the secret information enclosed in the regular binary frames. Remarkably, this technique increases quadratically the secret key rate as a function of the double matching detection events and doubly quadratically in the number of the quantum pulses. Furthermore, this reconciliation method opens up the opportunity to use less attenuated quantum pulses, would allow greater QKD distances at drastically increased secret key rate. Since our method can be implemented as a software update, we hope that quantum key distribution technology would be fast deployed over global data networks in the quantum era.

**Keywords:** QKD; distillation; reconciliation





## 1. Introduction

The arrival of the quantum era and quantum computers in the short term is imminent. One of the most profound consequences of the quantum era is that the security of digital data as we know it today must be radically changed due to the power of computers to break the security of asymmetric key cryptographic methods and at the same time must increase the sizes of the symmetrical keys.

Countermeasures to address the threat of quantum computers have been led by NIST, which launched a selection process for new cryptographic algorithms for the quantum era in 2017 [1]. However, it is to be expected that it will take years to implement and technologically adapt the new methods to be used in global data networks [2].

Fortunately, the quantum cryptographic key distribution (QKD) is a cryptographic scheme that appeared almost four decades ago that has been widely evaluated and discussed by the scientific community. In addition, QKD can be implemented through satellite links or already installed fiber optic networks. Quantum Key Distribution (QKD) allows the establishment of a secret key between two remote entities taking advantage of the principles of quantum physics and therefore theoretically secure. Unfortunately, the transmission of quantum states through a noisy quantum channel causes errors to appear in the received information, severely limiting the deployment of QKD technology as it reduces the secret bit rate and the distance achievable by the QKD system. For this reason, error correction algorithms have been developed to detect and correct errors during the post-processing phase, which includes sifting, error reconciliation, and privacy amplification. The reconciliation algorithm must be carried out preserving the secrecy of the cryptographic key [3].

Reconciliation methods developed natively for QKD are BBBSS [4] based on binary search, Cascade [5,6] that uses binary search and backtracking, but they are based on the parity computation of the received information blocks and do not take advantage of the properties of communication with quantum states; instead, it is highly interactive, requiring multiple rounds of correction of bits.

In view of the above, it has been necessary to resort to other reconciliation techniques developed in the field of data communications. Reconciliation methods based on error correcting codes are Winnow [7–9] which uses parity check and Hamming error correction code. It corrects one error per block, so the choice of block length is very sensitive because additional errors may be introduced if a block contains two or more errors [10]. Also, Forward Error Correction (FEC) is used to achieve reconciliation as the discrete number of Low-Density Parity-Check (LDPC) codes. However, LDPC has the disadvantage that requires redundant information that must be transmitted along the information data [11–15]. Recently, polar encoding has emerged as an encoding method in finding error correction codes that are close to the Shannon limit [16–18]. Beyond the mentioned drawbacks, none of these schemes is capable of handling a quantum channel error rate beyond 25% [19].

We published in [20,21] a new reconciliation algorithm that takes advantage of the characteristics of quantum communication, which, simply put, is equivalent to having two classical communication channels, one in the quantum **X** basis and the other in the **Z** basis. By means of a reverse reconciliation process, Bob sends parity information from these two channels so that Alice is able to recognize Bob's chosen bases on which the secret information is encoded as depicted in Figure 1.

**Alice's frame**  **Bob's frame**

$$\begin{pmatrix} 0_X & 1_Z \\ 1_X & 1_Z \end{pmatrix} \longrightarrow \begin{pmatrix} 0_X & + \\ + & 1_Z \end{pmatrix}$$

**Figure 1.** General scheme of frame-based QKD approach. Each row of the frame is a pair of non-orthogonal states: each position inside the frame encodes a quantum bit, so in the first position is $|i_X\rangle$ or $|(1-i)_X\rangle$ while in the second $|i_Z\rangle$ or $|(1-i)_Z\rangle$ where $i = 0, 1$. The symbol $+$ denotes an empty state.

A two order frame, as it can be seen in Figure 1, is a $2 \times 2$ matrix, structured by two rows and two columns. One row represents a pair of non-orthogonal states where the first column of the frame contains the base **X** encoded quantum bit $|i_X\rangle$ or $|(1-i)_X\rangle$ and the second column the base **Z** quantum bit, that is $|i_Z\rangle$ or $|(1-i)_Z\rangle$ where $i = 0, 1$. Once Bob measures a pair of non-orthogonal states and provided he gets a DMDE (Double Matching Detection Event), he obtains the encoded bit in the first or second column of the first row of the frame, as shown in Figures 1 and 2. Bob's frame is complete once he gets the second DMDE.

In fact, the two rows of a frame are not received sequentially, instead Bob must inform Alice about the DMDE that he obtained, then Alice responds to Bob how to arrange the rows to construct the frames as illustrated in the message exchange in Figure 3. For this reason, each DMDE is labelled by an index that has the form (CSS, $i_1$, $i_2$) where $i_1$ is the number of the first NO-QP and $i_2$ is the number of the second NO-QP. The index is assigned during transmission, so is known by Alice and Bob, CSS will be explained shortly.

This article is focused on the discussion and explanation of the reconciliation method, rather than on a detailed discussion of the attacks over the system. So, Section 2 contains a detailed discussion of the reconciliation method. In Section 3 we derive the relation for the secret throughput of the frame-based method. Finally, Section 4 contains a brief discussion about the main quantum attacks. But before going to the details of the reconciliation method, let us succinctly state the research problem and the general idea of the new method.

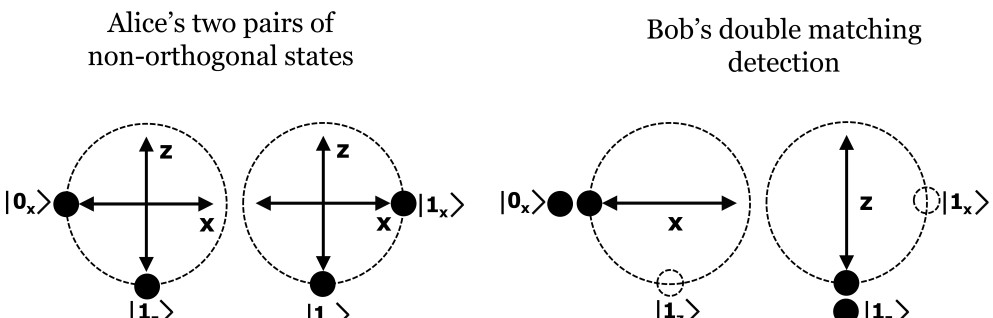

**Figure 2.** Alice sends two pairs of non-orthogonal states to Bob over the quantum channel: $(|0_X\rangle, |1_Z\rangle)$ and $(|1_X\rangle, |1_Z\rangle)$. Bob's gets two events of double matching detection events: $(|0_X\rangle, +)$ and $(+, |1_Z\rangle)$.

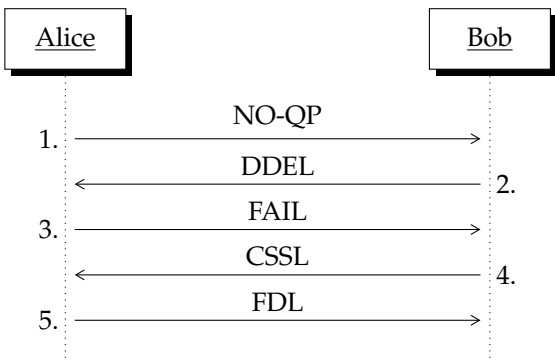

**Figure 3.** The reconciliation message exchange: NO-QP represents the quantum pulses over the quantum channel. Remaining steps take place over the classical channel. The meaning of the acronyms used is shown below. NO-QP: Non-Orthogonal Quantum Pairs. DDEL: the Double Detection Event List. FAIL: the Frame Arrangement Information List. CSSL: the Composed Sifting String List. FDL: the Frame to Delete List.

### 1.1. Research Problem Statement

The reconciliation method must be able to detect the errors produced into the pairs of non-orthogonal quantum states $\left(\overline{0_X}, 0_Z\right)$, $\left(0_X, \overline{0_Z}\right)$, $\left(\overline{0_X}, 1_Z\right)$, $\left(0_X, \overline{1_Z}\right)$, $\left(\overline{1_X}, 0_Z\right)$, $\left(1_X, \overline{0_Z}\right)$, $\left(\overline{1_X}, 1_Z\right)$ and $\left(1_X, \overline{1_Z}\right)$ where the overbracket symbol ⊓ represents the error that is produced in a transmitted NO-QP. We argue that if we can detect all these types of errors, we will achieve error correction that is invariant with respect to the error rate of the quantum channel. The following types of frames (which have been enumerated according to [20]) will be used:

1.  Auxiliary frames, of two types, null and unitary frames as well as their conjugate denoted with a math apostrophe. We call $f_7$ the null frame while $f_{11}$ is the unitary frame:

$$f_7 = \begin{pmatrix} 0_X & 0_Z \\ 0_X & 0_Z \end{pmatrix} \quad f_7{}' = \begin{pmatrix} 1_X & 1_Z \\ 1_X & 1_Z \end{pmatrix} \quad f_{11} = \begin{pmatrix} 1_X & 1_Z \\ 1_X & 1_Z \end{pmatrix} \quad f_{11}{}' = \begin{pmatrix} 0_X & 0_Z \\ 0_X & 0_Z \end{pmatrix}$$

2.  Regular-frames (twelve types) and their conjugate:

$$f_1 = \begin{pmatrix} 0_X & 1_Z \\ 1_X & 0_Z \end{pmatrix} \quad f_1{}' = \begin{pmatrix} 1_X & 0_Z \\ 0_X & 1_Z \end{pmatrix} \quad f_5 = \begin{pmatrix} 1_X & 0_Z \\ 0_X & 1_Z \end{pmatrix} \quad f_5{}' = \begin{pmatrix} 0_X & 1_Z \\ 1_X & 0_Z \end{pmatrix}$$

$$f_2 = \begin{pmatrix} 1_X & 0_Z \\ 1_X & 1_Z \end{pmatrix} \quad f_2{}' = \begin{pmatrix} 0_X & 1_Z \\ 0_X & 0_Z \end{pmatrix} \quad f_6 = \begin{pmatrix} 1_X & 1_Z \\ 1_X & 0_Z \end{pmatrix} \quad f_6{}' = \begin{pmatrix} 0_X & 0_Z \\ 0_X & 1_Z \end{pmatrix}$$

$$f_3 = \begin{pmatrix} \mathbf{0_X} & \mathbf{1_Z} \\ \mathbf{1_X} & \mathbf{1_Z} \end{pmatrix} \quad f_3{}' = \begin{pmatrix} \mathbf{1_X} & \mathbf{0_Z} \\ \mathbf{0_X} & \mathbf{0_Z} \end{pmatrix} \quad f_4 = \begin{pmatrix} \mathbf{1_X} & \mathbf{1_Z} \\ \mathbf{0_X} & \mathbf{1_Z} \end{pmatrix} \quad f_4{}' = \begin{pmatrix} \mathbf{0_X} & \mathbf{0_Z} \\ \mathbf{1_X} & \mathbf{0_Z} \end{pmatrix}$$

$$f_9 = \begin{pmatrix} \mathbf{0_X} & \mathbf{1_Z} \\ \mathbf{0_X} & \mathbf{0_Z} \end{pmatrix} \quad f_9{}' = \begin{pmatrix} \mathbf{1_X} & \mathbf{0_Z} \\ \mathbf{1_X} & \mathbf{1_Z} \end{pmatrix} \quad f_{10} = \begin{pmatrix} \mathbf{1_X} & \mathbf{0_Z} \\ \mathbf{0_X} & \mathbf{0_Z} \end{pmatrix} \quad f_{10}{}' = \begin{pmatrix} \mathbf{0_X} & \mathbf{1_Z} \\ \mathbf{1_X} & \mathbf{1_Z} \end{pmatrix}$$

$$f_8 = \begin{pmatrix} \mathbf{0_X} & \mathbf{0_Z} \\ \mathbf{1_X} & \mathbf{1_Z} \end{pmatrix} \quad f_8 = \begin{pmatrix} \mathbf{1_X} & \mathbf{1_Z} \\ \mathbf{0_X} & \mathbf{0_Z} \end{pmatrix} \quad f_{12} = \begin{pmatrix} \mathbf{1_X} & \mathbf{1_Z} \\ \mathbf{0_X} & \mathbf{0_Z} \end{pmatrix} \quad f_{12}{}' = \begin{pmatrix} \mathbf{0_X} & \mathbf{0_Z} \\ \mathbf{1_X} & \mathbf{1_Z} \end{pmatrix}$$

$$f_{13} = \begin{pmatrix} \mathbf{0_X} & \mathbf{0_Z} \\ \mathbf{0_X} & \mathbf{1_Z} \end{pmatrix} \quad f_{13} = \begin{pmatrix} \mathbf{1_X} & \mathbf{1_Z} \\ \mathbf{1_X} & \mathbf{0_Z} \end{pmatrix} \quad f_{14} = \begin{pmatrix} \mathbf{0_X} & \mathbf{0_Z} \\ \mathbf{1_X} & \mathbf{0_Z} \end{pmatrix} \quad f_{14}{}' = \begin{pmatrix} \mathbf{1_X} & \mathbf{1_Z} \\ \mathbf{0_X} & \mathbf{1_Z} \end{pmatrix}$$

The frame $f_{15} = \begin{pmatrix} \mathbf{0_X} & \mathbf{1_Z} \\ \mathbf{0_X} & \mathbf{1_Z} \end{pmatrix}$ and the frame $f_{16} = \begin{pmatrix} \mathbf{1_X} & \mathbf{0_Z} \\ \mathbf{1_X} & \mathbf{0_Z} \end{pmatrix}$ are not used in this context. Bob obtains the conjugate frames by inverting the measured bits so they are not obtained from the channel measurements.

### 1.2. General Idea of the Method

It is our goal to demonstrate that using the Composed Sifting String (CSS) is possible that Alice reconciliate 100% of the errors produced in received Bob's DMDE.

— Just to bring it in context, the Sifting String (SS) as stated in [20] is constructed using the sifting bits and the measured bits into Bob's frames. The sifting bits are obtained applying the XOR function to the bits within the columns (from the left to the right column) of Bob's frames, where a vacuum bit is taken as a zero bit. The measured bits are taken directly from the bits inside Bob's frame. This is so because the secret bit is derived from the final configuration of Bob's frames, that we call Measurement Results (MR) as represented in Table 1. The sifting bits are written first into SS while the measured bits are placed next:

$$\text{SS} = \text{1st sifting bit} \parallel \text{2nd sifting bit}, \ \text{1st measured bit} \parallel \text{2nd measured bit}$$

Unfortunately, using SS as designed in [20] is impossible to detect the errors $\left(\mathbf{0_X}, \overline{\mathbf{1_Z}}\right)$, $\left(\overline{\mathbf{1_X}}, \mathbf{0_Z}\right)$, $\left(\overline{\mathbf{1_X}}, \mathbf{1_Z}\right)$ and $\left(\mathbf{1_X}, \overline{\mathbf{1_Z}}\right)$.

— Now, in this new reconciliation method we introduce the Composed Sifting String (CSS) which is constructed taken the sifting bits of Bob's frame but also the sifting bits of Bob's conjugate frame, that is:

$$\text{CSS} = \text{1st sifting bit} \parallel \text{2nd sifting bit} \parallel$$
$$\text{1st sifting bit of conjugate frame} \parallel \text{2nd sifting bit of conjugate frame}$$

We will demonstrate that using CSS and without compromising the security of the scheme, is possible to detect the errors $\left(\overline{\mathbf{0_X}}, \mathbf{0_Z}\right)$, $\left(\mathbf{0_X}, \overline{\mathbf{0_Z}}\right)$, $\left(\overline{\mathbf{0_X}}, \mathbf{1_Z}\right)$, $\left(\mathbf{0_X}, \overline{\mathbf{1_Z}}\right)$, $\left(\overline{\mathbf{1_X}}, \mathbf{0_Z}\right)$, $\left(\mathbf{1_X}, \overline{\mathbf{0_Z}}\right)$, $\left(\overline{\mathbf{1_X}}, \mathbf{1_Z}\right)$ and $\left(\mathbf{1_X}, \overline{\mathbf{1_Z}}\right)$.

**Table 1.** Matching Results Table (MRT) for $2 \times 2$ frames. Since the bit that results from the measurement is not relevant to define MR but the position in which it appears, we use the symbol $\bullet$ to represent a quantum measurement while $+$ denotes an empty state.

| | | | |
|---|---|---|---|
| MR = 00 $\begin{pmatrix} \lvert\bullet_X\rangle & + \\ \lvert\bullet_X\rangle & + \end{pmatrix}$ | | MR = 10 $\begin{pmatrix} \lvert\bullet_X\rangle & + \\ + & \lvert\bullet_Z\rangle \end{pmatrix}$ | |
| MR = 01 $\begin{pmatrix} + & \lvert\bullet_Z\rangle \\ + & \lvert\bullet_Z\rangle \end{pmatrix}$ | | MR = 11 $\begin{pmatrix} + & \lvert\bullet_Z\rangle \\ \lvert\bullet_X\rangle & + \end{pmatrix}$ | |

## 2. Perfect Reconciliation Using Order-Two Binary Frames

We begin this section by establishing the general steps of the reconciliation method which we will justify throughout the section. To simplify notation, throughout this document we will represent a quantum state using a bold letter instead of the usual ket notation, so we denote $|i_X\rangle$ as $\mathbf{i_X}$.

1.  Alice creates NO-QPL (the Non-Orthogonal Quantum Pair List) and sends, one by one, each NO-QP (the Non-Orthogonal Quantum Pair) across QC (the Quantum Channel). NO-QP can be $(\mathbf{i_X}, \mathbf{i_Z})$ or $(\mathbf{i_X}, (\mathbf{1-i})_\mathbf{Z})$ where $i = 0, 1$.
2.  Bob chooses randomly the measurement basis: **X** or **Z**, that he will use to measure both states inside NO-QP. After Bob registers DDE (the Double Detection Events) he sends DDEL (the Double Detection Event List) to Alice.
3.  Alice receive DDEL from QC, she creates FAIL (the Frame Arrangement Information List) and sends it to Bob.
4.  Bob receives FAIL and he computes CSSL (the Composed Sifting String List). Then he returns CSSL to Alice.
5.  Alice detect errors and identifies MR in regular frames. Alice sends FDL (the Frame to Delete List) to Bob.
6.  Bob eliminates the frames indicated in FDL, then he creates SeS using MRT as written in Table 1.

We call step 3 of the protocol privacy pre-amplification, in this step Alice performs all the combinations of the DMDE to form the frames that she is going to use in order to successfully carry out the error correction process. Then, the number of possible frames is given by the combination formula $\binom{n}{2} = \frac{n!}{2!(n-2)!}$ where $n$ is the number of DMDE. The general diagram showing the protocol message exchange is shown in Figure 3.

### 2.1. Regular and Conjugate 2 × 2 Frames

Conjugate frames are derived from regular frames thus they are not obtained from the physical quantum channel. They are used just to derive a useful complementary set of sifting bits. Below we will show each one of the regular frames, each one with its respective conjugated frame and we will add its corresponding CSS each MR.

$$
\begin{pmatrix} \mathbf{0_X} & \mathbf{1_Z} \\ \mathbf{1_X} & \mathbf{0_Z} \end{pmatrix} : \underbrace{\begin{pmatrix} + & \mathbf{1_Z} \\ + & \mathbf{0_Z} \end{pmatrix}}_{01} \underbrace{\begin{pmatrix} \mathbf{0_X} & + \\ \mathbf{1_X} & + \end{pmatrix}}_{10} \underbrace{\begin{pmatrix} \mathbf{0_X} & + \\ + & \mathbf{0_Z} \end{pmatrix}}_{00} \underbrace{\begin{pmatrix} + & \mathbf{1_Z} \\ \mathbf{1_X} & + \end{pmatrix}}_{11}
$$

$$
\begin{pmatrix} \mathbf{1_X} & \mathbf{0_Z} \\ \mathbf{0_X} & \mathbf{1_Z} \end{pmatrix} : \underbrace{\begin{pmatrix} + & \mathbf{0_Z} \\ + & \mathbf{1_Z} \end{pmatrix}}_{01} \underbrace{\begin{pmatrix} \mathbf{1_X} & + \\ \mathbf{0_X} & + \end{pmatrix}}_{10} \underbrace{\begin{pmatrix} \mathbf{1_X} & + \\ + & \mathbf{1_Z} \end{pmatrix}}_{11} \underbrace{\begin{pmatrix} + & \mathbf{0_Z} \\ \mathbf{0_X} & + \end{pmatrix}}_{00}
$$

$$
\text{CSS}: \quad 0101 \quad 1010 \quad 0011 \quad 1100 \tag{$f_1$}
$$

$$
\begin{pmatrix} \mathbf{1_X} & \mathbf{0_Z} \\ \mathbf{1_X} & \mathbf{1_Z} \end{pmatrix} : \underbrace{\begin{pmatrix} + & \mathbf{0_Z} \\ + & \mathbf{1_Z} \end{pmatrix}}_{01} \underbrace{\begin{pmatrix} \mathbf{1_X} & + \\ \mathbf{1_X} & + \end{pmatrix}}_{00} \underbrace{\begin{pmatrix} \mathbf{1_X} & + \\ + & \mathbf{1_Z} \end{pmatrix}}_{11} \underbrace{\begin{pmatrix} + & \mathbf{0_Z} \\ \mathbf{1_X} & + \end{pmatrix}}_{10}
$$

$$
\begin{pmatrix} \mathbf{0_X} & \mathbf{1_Z} \\ \mathbf{0_X} & \mathbf{0_Z} \end{pmatrix} : \underbrace{\begin{pmatrix} + & \mathbf{1_Z} \\ + & \mathbf{0_Z} \end{pmatrix}}_{01} \underbrace{\begin{pmatrix} \mathbf{0_X} & + \\ \mathbf{0_X} & + \end{pmatrix}}_{00} \underbrace{\begin{pmatrix} \mathbf{0_X} & + \\ + & \mathbf{0_Z} \end{pmatrix}}_{00} \underbrace{\begin{pmatrix} + & \mathbf{1_Z} \\ \mathbf{0_X} & + \end{pmatrix}}_{01}
$$

$$
\text{CSS}: \quad 0101 \quad 0000 \quad 1100 \quad 1001 \tag{$f_2$}
$$

$$\begin{pmatrix} \mathbf{0_X} & \mathbf{1_Z} \\ \mathbf{1_X} & \mathbf{1_Z} \end{pmatrix} : \underbrace{\begin{pmatrix} + & \mathbf{1_Z} \\ + & \mathbf{1_Z} \end{pmatrix}}_{00} \underbrace{\begin{pmatrix} \mathbf{0_X} & + \\ \mathbf{1_X} & + \end{pmatrix}}_{10} \underbrace{\begin{pmatrix} \mathbf{0_X} & + \\ + & \mathbf{1_Z} \end{pmatrix}}_{01} \underbrace{\begin{pmatrix} + & \mathbf{1_Z} \\ \mathbf{1_X} & + \end{pmatrix}}_{11}$$

$$\begin{pmatrix} \mathbf{1_X} & \mathbf{0_Z} \\ \mathbf{0_X} & \mathbf{0_Z} \end{pmatrix} : \underbrace{\begin{pmatrix} + & \mathbf{0_Z} \\ + & \mathbf{0_Z} \end{pmatrix}}_{00} \underbrace{\begin{pmatrix} \mathbf{1_X} & + \\ \mathbf{0_X} & + \end{pmatrix}}_{10} \underbrace{\begin{pmatrix} \mathbf{1_X} & + \\ + & \mathbf{0_Z} \end{pmatrix}}_{10} \underbrace{\begin{pmatrix} + & \mathbf{0_Z} \\ \mathbf{0_X} & + \end{pmatrix}}_{00}$$

$$\text{CSS}: \quad 0000 \quad 1010 \quad 0110 \quad 1100 \tag{$f_3$}$$

$$\begin{pmatrix} \mathbf{1_X} & \mathbf{1_Z} \\ \mathbf{0_X} & \mathbf{1_Z} \end{pmatrix} : \underbrace{\begin{pmatrix} + & \mathbf{1_Z} \\ + & \mathbf{1_Z} \end{pmatrix}}_{00} \underbrace{\begin{pmatrix} \mathbf{1_X} & + \\ \mathbf{0_X} & + \end{pmatrix}}_{10} \underbrace{\begin{pmatrix} \mathbf{1_X} & + \\ + & \mathbf{1_Z} \end{pmatrix}}_{11} \underbrace{\begin{pmatrix} + & \mathbf{1_Z} \\ \mathbf{0_X} & + \end{pmatrix}}_{01}$$

$$\begin{pmatrix} \mathbf{0_X} & \mathbf{0_Z} \\ \mathbf{1_X} & \mathbf{0_Z} \end{pmatrix} : \underbrace{\begin{pmatrix} + & \mathbf{0_Z} \\ + & \mathbf{0_Z} \end{pmatrix}}_{00} \underbrace{\begin{pmatrix} \mathbf{0_X} & + \\ \mathbf{1_X} & + \end{pmatrix}}_{10} \underbrace{\begin{pmatrix} \mathbf{0_X} & + \\ + & \mathbf{0_Z} \end{pmatrix}}_{00} \underbrace{\begin{pmatrix} + & \mathbf{0_Z} \\ \mathbf{1_X} & + \end{pmatrix}}_{10}$$

$$\text{CSS}: \quad 0000 \quad 1010 \quad 1100 \quad 0110 \tag{$f_4$}$$

$$\begin{pmatrix} \mathbf{1_X} & \mathbf{0_Z} \\ \mathbf{0_X} & \mathbf{1_Z} \end{pmatrix} : \underbrace{\begin{pmatrix} + & \mathbf{0_Z} \\ + & \mathbf{1_Z} \end{pmatrix}}_{01} \underbrace{\begin{pmatrix} \mathbf{1_X} & + \\ \mathbf{0_X} & + \end{pmatrix}}_{10} \underbrace{\begin{pmatrix} \mathbf{1_X} & + \\ + & \mathbf{1_Z} \end{pmatrix}}_{11} \underbrace{\begin{pmatrix} + & \mathbf{0_Z} \\ \mathbf{0_X} & + \end{pmatrix}}_{00}$$

$$\begin{pmatrix} \mathbf{0_X} & \mathbf{1_Z} \\ \mathbf{1_X} & \mathbf{0_Z} \end{pmatrix} : \underbrace{\begin{pmatrix} + & \mathbf{1_Z} \\ + & \mathbf{0_Z} \end{pmatrix}}_{01} \underbrace{\begin{pmatrix} \mathbf{0_X} & + \\ \mathbf{1_X} & + \end{pmatrix}}_{10} \underbrace{\begin{pmatrix} \mathbf{0_X} & + \\ + & \mathbf{0_Z} \end{pmatrix}}_{00} \underbrace{\begin{pmatrix} + & \mathbf{1_Z} \\ \mathbf{1_X} & + \end{pmatrix}}_{11}$$

$$\text{CSS}: \quad 0101 \quad 1010 \quad 1100 \quad 0011 \tag{$f_5$}$$

$$\begin{pmatrix} \mathbf{1_X} & \mathbf{1_Z} \\ \mathbf{1_X} & \mathbf{0_Z} \end{pmatrix} : \underbrace{\begin{pmatrix} + & \mathbf{1_Z} \\ + & \mathbf{0_Z} \end{pmatrix}}_{01} \underbrace{\begin{pmatrix} \mathbf{1_X} & + \\ \mathbf{1_X} & + \end{pmatrix}}_{00} \underbrace{\begin{pmatrix} \mathbf{1_X} & + \\ + & \mathbf{0_Z} \end{pmatrix}}_{10} \underbrace{\begin{pmatrix} + & \mathbf{1_Z} \\ \mathbf{1_X} & + \end{pmatrix}}_{11}$$

$$\begin{pmatrix} \mathbf{0_X} & \mathbf{0_Z} \\ \mathbf{0_X} & \mathbf{1_Z} \end{pmatrix} : \underbrace{\begin{pmatrix} + & \mathbf{0_Z} \\ + & \mathbf{1_Z} \end{pmatrix}}_{01} \underbrace{\begin{pmatrix} \mathbf{0_X} & + \\ \mathbf{0_X} & + \end{pmatrix}}_{00} \underbrace{\begin{pmatrix} \mathbf{0_X} & + \\ + & \mathbf{1_Z} \end{pmatrix}}_{01} \underbrace{\begin{pmatrix} + & \mathbf{0_Z} \\ \mathbf{0_X} & + \end{pmatrix}}_{00}$$

$$\text{CSS}: \quad 0101 \quad 0000 \quad 1001 \quad 1100 \tag{$f_6$}$$

The sifting algorithm can also be applied to the remaining regular frames $f_8$, $f_{12}$, $f_9$, $f_{10}$, $f_{14}$, $f_{14}$ which we do not show here to facilitate the exposition of the method. The security property or frame-based model states that each CSS must map to at least two MR because the secret bit is derived from MR. The results are presented in Table 2. As can be seen there, the cases CSS 1010 and 0101 should be removed because they map a single MR: 00 and 01, respectively.

**Table 2.** Each Composed Sifting String (CSS) must be correlated at least to two Matching Results (MR). The symbol sb denotes the secret bit.

| CSS | Frame | MR | Sb | Action |
|:---:|:---:|:---:|:---:|:---:|
| 0110 | $f_3, f_8, f_{13}$ | 10 | 0 | distill |
|  | $f_4, f_{12}, f_9$ | 11 | 1 |  |
| 1001 | $f_2, f_8, f_{14}$ | 11 | 0 | distill |
|  | $f_6, f_{12}, f_{10}$ | 10 | 1 |  |
| 0011 | $f_1, f_9, f_{14}$ | 10 | 0 | distill |
|  | $f_5, f_{10}, f_{13}$ | 11 | 1 |  |
| 0000 | $f_2, f_6, f_9, f_{13}$ | 00 | 0 | distill |
|  | $f_3, f_4, f_{10}, f_{14}$ | 01 | 1 |  |
| 1100 | $f_1, f_3, f_6$ | 11 | 0 | distill |
|  | $f_2, f_4, f_5$ | 10 | 1 |  |
| 1010 | $f_1, f_3, f_4, f_5$ | 00 | - | remove |
|  | $f_8, f_{12}, f_{10}, f_{14}$ |  |  |  |
| 0101 | $f_1, f_2, f_5, f_6$ | 01 | - | remove |
|  | $f_8, f_{12}, f_9, f_{13}$ |  |  |  |

Now, we proceed to demonstrate which errors can be detected inside a frame. Due to their structure, is convenient to see the frames grouped as: $\{f_1, f_5\}$, $\{f_3, f_4\}$, $\{f_2, f_6\}$, $\{f_9, f_{10}\}$, $\{f_8, f_{12}\}$. In the following equations, the top line contains the frame under MR while the second line shows the conjugated frame. The bottom line hold the computed CSS in each case. The error detection illustrated depends on an error-free NO-QP, so we will solve this point as the error correction pre-processing.

1. The error $\left(\boxed{1_X}, 0_Z\right)$ is detected with an error-free NO-QP $(1_X, 1_Z)$, as the second row in $f_2$ because the CSS that is produced by the error is none of the possible error-free CSS.

$$
\begin{pmatrix} 1_X & 0_Z \\ 1_X & 1_Z \end{pmatrix} : \underset{01}{\begin{pmatrix} + & 0_Z \\ + & 1_Z \end{pmatrix}} \underset{00}{\begin{pmatrix} 1_X & + \\ 1_X & + \end{pmatrix}} \underset{11}{\begin{pmatrix} 1_X & + \\ + & 1_Z \end{pmatrix}} \underset{10}{\begin{pmatrix} + & 0_Z \\ 1_X & + \end{pmatrix}}
$$

$$
\begin{pmatrix} 0_X & 1_Z \\ 0_X & 0_Z \end{pmatrix} : \underset{01}{\begin{pmatrix} + & 1_Z \\ + & 0_Z \end{pmatrix}} \underset{00}{\begin{pmatrix} 0_X & + \\ 0_X & + \end{pmatrix}} \underset{00}{\begin{pmatrix} 0_X & + \\ + & 0_Z \end{pmatrix}} \underset{01}{\begin{pmatrix} + & 1_Z \\ 0_X & + \end{pmatrix}}
$$

$$
\text{CSS}: \quad 0101 \quad 0000 \quad 1100 \quad 1001
$$

$$
\begin{pmatrix} \boxed{1_X} & 0_Z \\ 1_X & 1_Z \end{pmatrix} : \underset{01}{\begin{pmatrix} + & 0_Z \\ + & 1_Z \end{pmatrix}} \underset{10}{\begin{pmatrix} \boxed{0_X} & + \\ 1_X & + \end{pmatrix}} \underset{01}{\begin{pmatrix} \boxed{0_X} & + \\ + & 1_Z \end{pmatrix}} \underset{10}{\begin{pmatrix} + & 0_Z \\ 1_X & + \end{pmatrix}}
$$

$$
\begin{pmatrix} \boxed{0_X} & 1_Z \\ 0_X & 0_Z \end{pmatrix} : \underset{01}{\begin{pmatrix} + & 1_Z \\ + & 0_Z \end{pmatrix}} \underset{10}{\begin{pmatrix} \boxed{1_X} & + \\ 0_X & + \end{pmatrix}} \underset{10}{\begin{pmatrix} \boxed{1_X} & + \\ + & 0_Z \end{pmatrix}} \underset{01}{\begin{pmatrix} + & 1_Z \\ 0_X & + \end{pmatrix}}
$$

$$
\text{CSS}: \quad 0101 \quad \underline{1010} \quad \underline{0110} \quad 1001 \qquad (f_2)
$$

2. The error $\left(\boxed{1_X}, 0_Z\right)$ is detected with an error-free NO-QP $(1_X, 1_Z)$, as the first row in $f_6$ because the CSS that is produced by the error is none of the possible error-free CSS.

$$\begin{pmatrix} \mathbf{1_X} & \mathbf{1_Z} \\ \mathbf{1_X} & \mathbf{0_Z} \end{pmatrix} : \underbrace{\begin{pmatrix} + & \mathbf{1_Z} \\ + & \mathbf{0_Z} \end{pmatrix}}_{01} \underbrace{\begin{pmatrix} \mathbf{1_X} & + \\ \mathbf{1_X} & + \end{pmatrix}}_{00} \underbrace{\begin{pmatrix} \mathbf{1_X} & + \\ + & \mathbf{0_Z} \end{pmatrix}}_{10} \underbrace{\begin{pmatrix} + & \mathbf{1_Z} \\ \mathbf{1_X} & + \end{pmatrix}}_{11}$$

$$\begin{pmatrix} \mathbf{0_X} & \mathbf{0_Z} \\ \mathbf{0_X} & \mathbf{1_Z} \end{pmatrix} : \underbrace{\begin{pmatrix} + & \mathbf{0_Z} \\ + & \mathbf{1_Z} \end{pmatrix}}_{01} \underbrace{\begin{pmatrix} \mathbf{0_X} & + \\ \mathbf{0_X} & + \end{pmatrix}}_{00} \underbrace{\begin{pmatrix} \mathbf{0_X} & + \\ + & \mathbf{1_Z} \end{pmatrix}}_{01} \underbrace{\begin{pmatrix} + & \mathbf{0_Z} \\ \mathbf{0_X} & + \end{pmatrix}}_{00}$$

$$\text{CSS}: \quad 0101 \quad 0000 \quad 1001 \quad 1100$$

$$\begin{pmatrix} \mathbf{1_X} & \mathbf{1_Z} \\ \boxed{\mathbf{1_X}} & \mathbf{0_Z} \end{pmatrix} : \underbrace{\begin{pmatrix} + & \mathbf{1_Z} \\ + & \mathbf{0_Z} \end{pmatrix}}_{01} \underbrace{\begin{pmatrix} \mathbf{1_X} & + \\ \boxed{\mathbf{0_X}} & + \end{pmatrix}}_{10} \underbrace{\begin{pmatrix} \mathbf{1_X} & + \\ + & \mathbf{0_Z} \end{pmatrix}}_{10} \underbrace{\begin{pmatrix} + & \mathbf{1_Z} \\ \boxed{\mathbf{0_X}} & + \end{pmatrix}}_{01}$$

$$\begin{pmatrix} \mathbf{0_X} & \mathbf{0_Z} \\ \boxed{\mathbf{0_X}} & \mathbf{1_Z} \end{pmatrix} : \underbrace{\begin{pmatrix} + & \mathbf{0_Z} \\ + & \mathbf{1_Z} \end{pmatrix}}_{01} \underbrace{\begin{pmatrix} \mathbf{0_X} & + \\ \boxed{\mathbf{1_X}} & + \end{pmatrix}}_{10} \underbrace{\begin{pmatrix} \mathbf{0_X} & + \\ + & \mathbf{1_Z} \end{pmatrix}}_{01} \underbrace{\begin{pmatrix} + & \mathbf{0_Z} \\ \boxed{\mathbf{1_X}} & + \end{pmatrix}}_{10}$$

$$\text{CSS}: \quad 0101 \quad \underline{1010} \quad 1001 \quad \underline{0110} \qquad\qquad (f_6)$$

3. The error $\left( \mathbf{0_X}, \boxed{\mathbf{1_Z}} \right)$ is detected with an error-free NO-QP $(\mathbf{1_X}, \mathbf{1_Z})$, as the second row in $f_3$ because the CSS that is produced by the error is none of the possible error-free CSS.

$$\begin{pmatrix} \mathbf{0_X} & \mathbf{1_Z} \\ \mathbf{1_X} & \mathbf{1_Z} \end{pmatrix} : \underbrace{\begin{pmatrix} + & \mathbf{1_Z} \\ + & \mathbf{1_Z} \end{pmatrix}}_{00} \underbrace{\begin{pmatrix} \mathbf{0_X} & + \\ \mathbf{1_X} & + \end{pmatrix}}_{10} \underbrace{\begin{pmatrix} \mathbf{0_X} & + \\ + & \mathbf{1_Z} \end{pmatrix}}_{01} \underbrace{\begin{pmatrix} + & \mathbf{1_Z} \\ \mathbf{1_X} & + \end{pmatrix}}_{11}$$

$$\begin{pmatrix} \mathbf{1_X} & \mathbf{0_Z} \\ \mathbf{0_X} & \mathbf{0_Z} \end{pmatrix} : \underbrace{\begin{pmatrix} + & \mathbf{0_Z} \\ + & \mathbf{0_Z} \end{pmatrix}}_{00} \underbrace{\begin{pmatrix} \mathbf{1_X} & + \\ \mathbf{0_X} & + \end{pmatrix}}_{10} \underbrace{\begin{pmatrix} \mathbf{1_X} & + \\ + & \mathbf{0_Z} \end{pmatrix}}_{10} \underbrace{\begin{pmatrix} + & \mathbf{0_Z} \\ \mathbf{0_X} & + \end{pmatrix}}_{00}$$

$$\text{CSS}: \quad 0000 \quad 1010 \quad 0110 \quad 1100$$

$$\begin{pmatrix} \mathbf{0_X} & \boxed{\mathbf{1_Z}} \\ \mathbf{1_X} & \mathbf{1_Z} \end{pmatrix} : \underbrace{\begin{pmatrix} + & \boxed{\mathbf{0_Z}} \\ + & \mathbf{1_Z} \end{pmatrix}}_{01} \underbrace{\begin{pmatrix} \mathbf{0_X} & + \\ \mathbf{1_X} & + \end{pmatrix}}_{10} \underbrace{\begin{pmatrix} \mathbf{0_X} & + \\ + & \mathbf{1_Z} \end{pmatrix}}_{01} \underbrace{\begin{pmatrix} + & \boxed{\mathbf{0_Z}} \\ \mathbf{1_X} & + \end{pmatrix}}_{10}$$

$$\begin{pmatrix} \mathbf{1_X} & \boxed{\mathbf{0_Z}} \\ \mathbf{0_X} & \mathbf{0_Z} \end{pmatrix} : \underbrace{\begin{pmatrix} + & \boxed{\mathbf{1_Z}} \\ + & \mathbf{0_Z} \end{pmatrix}}_{01} \underbrace{\begin{pmatrix} \mathbf{1_X} & + \\ \mathbf{0_X} & + \end{pmatrix}}_{10} \underbrace{\begin{pmatrix} \mathbf{1_X} & + \\ + & \mathbf{0_Z} \end{pmatrix}}_{10} \underbrace{\begin{pmatrix} + & \boxed{\mathbf{1_Z}} \\ \mathbf{0_X} & + \end{pmatrix}}_{01}$$

$$\text{CSS}: \quad \underline{0101} \quad 1010 \quad 0110 \quad \underline{1001} \qquad\qquad (f_3)$$

4. The error $\left( \mathbf{0_X}, \boxed{\mathbf{1_Z}} \right)$ is detected with an error-free NO-QP $(\mathbf{1_X}, \mathbf{1_Z})$, as the first row in $f_4$ because the CSS that is produced by the error is none of the possible error-free CSS.

$$\begin{pmatrix} \mathbf{1_X} & \mathbf{1_Z} \\ \mathbf{0_X} & \mathbf{1_Z} \end{pmatrix} : \begin{pmatrix} + & \mathbf{1_Z} \\ + & \mathbf{1_Z} \end{pmatrix}_{00} \begin{pmatrix} \mathbf{1_X} & + \\ \mathbf{0_X} & + \end{pmatrix}_{10} \begin{pmatrix} \mathbf{1_X} & + \\ + & \mathbf{1_Z} \end{pmatrix}_{11} \begin{pmatrix} + & \mathbf{1_Z} \\ \mathbf{0_X} & + \end{pmatrix}_{01}$$

$$\begin{pmatrix} \mathbf{0_X} & \mathbf{0_Z} \\ \mathbf{1_X} & \mathbf{0_Z} \end{pmatrix} : \begin{pmatrix} + & \mathbf{0_Z} \\ + & \mathbf{0_Z} \end{pmatrix}_{00} \begin{pmatrix} \mathbf{0_X} & + \\ \mathbf{1_X} & + \end{pmatrix}_{10} \begin{pmatrix} \mathbf{0_X} & + \\ + & \mathbf{0_Z} \end{pmatrix}_{00} \begin{pmatrix} + & \mathbf{0_Z} \\ \mathbf{1_X} & + \end{pmatrix}_{10}$$

$$\text{CSS}: \quad 0000 \quad 1010 \quad 1100 \quad 0110$$

$$\begin{pmatrix} \mathbf{1_X} & \mathbf{1_Z} \\ \mathbf{0_X} & \boxed{\mathbf{1_Z}} \end{pmatrix} : \begin{pmatrix} + & \mathbf{1_Z} \\ + & \boxed{\mathbf{0_Z}} \end{pmatrix}_{01} \begin{pmatrix} \mathbf{1_X} & + \\ \mathbf{0_X} & + \end{pmatrix}_{10} \begin{pmatrix} \mathbf{1_X} & + \\ + & \boxed{\mathbf{0_Z}} \end{pmatrix}_{10} \begin{pmatrix} + & \mathbf{1_Z} \\ \mathbf{0_X} & + \end{pmatrix}_{01}$$

$$\begin{pmatrix} \mathbf{0_X} & \mathbf{0_Z} \\ \mathbf{1_X} & \boxed{\mathbf{0_Z}} \end{pmatrix} : \begin{pmatrix} + & \mathbf{0_Z} \\ + & \boxed{\mathbf{1_Z}} \end{pmatrix}_{01} \begin{pmatrix} \mathbf{0_X} & + \\ \mathbf{1_X} & + \end{pmatrix}_{10} \begin{pmatrix} \mathbf{0_X} & + \\ + & \boxed{\mathbf{1_Z}} \end{pmatrix}_{01} \begin{pmatrix} + & \mathbf{0_Z} \\ \mathbf{1_X} & + \end{pmatrix}_{10}$$

$$\text{CSS}: \quad \underline{0101} \quad 1010 \quad \underline{1001} \quad 0110 \tag{$f_4$}$$

5.  The error $\left(\boxed{\mathbf{0_X}}, \mathbf{1_Z}\right)$ is detected with an error-free NO-QP $(\mathbf{0_X}, \mathbf{0_Z})$, as the second row in $f_9$ because the CSS that is produced by the error is none of the possible error-free CSS.

$$\begin{pmatrix} \mathbf{0_X} & \mathbf{1_Z} \\ \mathbf{0_X} & \mathbf{0_Z} \end{pmatrix} : \begin{pmatrix} + & \mathbf{1_Z} \\ + & \mathbf{0_Z} \end{pmatrix}_{01} \begin{pmatrix} \mathbf{0_X} & + \\ \mathbf{0_X} & + \end{pmatrix}_{00} \begin{pmatrix} \mathbf{0_X} & + \\ + & \mathbf{0_Z} \end{pmatrix}_{00} \begin{pmatrix} + & \mathbf{1_Z} \\ \mathbf{0_X} & + \end{pmatrix}_{01}$$

$$\begin{pmatrix} \mathbf{1_X} & \mathbf{0_Z} \\ \mathbf{1_X} & \mathbf{1_Z} \end{pmatrix} : \begin{pmatrix} + & \mathbf{0_Z} \\ + & \mathbf{1_Z} \end{pmatrix}_{01} \begin{pmatrix} \mathbf{1_X} & + \\ \mathbf{1_X} & + \end{pmatrix}_{00} \begin{pmatrix} \mathbf{1_X} & + \\ + & \mathbf{1_Z} \end{pmatrix}_{11} \begin{pmatrix} + & \mathbf{0_Z} \\ \mathbf{1_X} & + \end{pmatrix}_{10}$$

$$\text{CSS}: \quad 0101 \quad 0000 \quad 0011 \quad 0110$$

$$\begin{pmatrix} \boxed{\mathbf{0_X}} & \mathbf{1_Z} \\ \mathbf{0_X} & \mathbf{0_Z} \end{pmatrix} : \begin{pmatrix} + & \mathbf{1_Z} \\ + & \mathbf{0_Z} \end{pmatrix}_{01} \begin{pmatrix} \boxed{\mathbf{1_X}} & + \\ \mathbf{0_X} & + \end{pmatrix}_{10} \begin{pmatrix} \boxed{\mathbf{1_X}} & + \\ + & \mathbf{0_Z} \end{pmatrix}_{10} \begin{pmatrix} + & \mathbf{1_Z} \\ \mathbf{0_X} & + \end{pmatrix}_{01}$$

$$\begin{pmatrix} \boxed{\mathbf{1_X}} & \mathbf{0_Z} \\ \mathbf{1_X} & \mathbf{1_Z} \end{pmatrix} : \begin{pmatrix} + & \mathbf{0_Z} \\ + & \mathbf{1_Z} \end{pmatrix}_{01} \begin{pmatrix} \boxed{\mathbf{0_X}} & + \\ \mathbf{1_X} & + \end{pmatrix}_{10} \begin{pmatrix} \boxed{\mathbf{0_X}} & + \\ + & \mathbf{1_Z} \end{pmatrix}_{01} \begin{pmatrix} + & \mathbf{0_Z} \\ \mathbf{1_X} & + \end{pmatrix}_{10}$$

$$\text{CSS}: \quad 0101 \quad \underline{1010} \quad \underline{1001} \quad 0110 \tag{$f_9$}$$

6.  The error $\left(\mathbf{1_X}, \boxed{\mathbf{0_Z}}\right)$ is detected with an error-free NO-QP $(\mathbf{0_X}, \mathbf{0_Z})$, as the second row in $f_{10}$ because the CSS that is produced by the error is none of the possible error-free CSS.

$$\begin{pmatrix} \mathbf{1_X} & \mathbf{0_Z} \\ \mathbf{0_X} & \mathbf{0_Z} \end{pmatrix} : \begin{pmatrix} + & \mathbf{0_Z} \\ + & \mathbf{0_Z} \end{pmatrix}_{00} \begin{pmatrix} \mathbf{1_X} & + \\ \mathbf{0_X} & + \end{pmatrix}_{10} \begin{pmatrix} \mathbf{1_X} & + \\ + & \mathbf{0_Z} \end{pmatrix}_{10} \begin{pmatrix} + & \mathbf{0_Z} \\ \mathbf{0_X} & + \end{pmatrix}_{00}$$

$$\begin{pmatrix} \mathbf{0_X} & \mathbf{1_Z} \\ \mathbf{1_X} & \mathbf{1_Z} \end{pmatrix} : \begin{pmatrix} + & \mathbf{1_Z} \\ + & \mathbf{1_Z} \end{pmatrix}_{00} \begin{pmatrix} \mathbf{0_X} & + \\ \mathbf{1_X} & + \end{pmatrix}_{10} \begin{pmatrix} \mathbf{0_X} & + \\ + & \mathbf{1_Z} \end{pmatrix}_{01} \begin{pmatrix} + & \mathbf{1_Z} \\ \mathbf{1_X} & + \end{pmatrix}_{11}$$

$$\text{CSS}: \quad 0000 \quad 1010 \quad 1001 \quad 0011$$

$$\begin{pmatrix} \mathbf{1_X} & \overline{\mathbf{0_Z}} \\ \mathbf{0_X} & \mathbf{0_Z} \end{pmatrix} : \begin{pmatrix} + & \overline{\mathbf{1_Z}} \\ + & \mathbf{0_Z} \end{pmatrix}_{01} \begin{pmatrix} \mathbf{1_X} & + \\ \mathbf{0_X} & + \end{pmatrix}_{10} \begin{pmatrix} \mathbf{1_X} & + \\ + & \mathbf{0_Z} \end{pmatrix}_{10} \begin{pmatrix} + & \overline{\mathbf{1_Z}} \\ \mathbf{0_X} & + \end{pmatrix}_{01}$$

$$\begin{pmatrix} \mathbf{0_X} & \overline{\mathbf{1_Z}} \\ \mathbf{1_X} & \mathbf{1_Z} \end{pmatrix} : \begin{pmatrix} + & \overline{\mathbf{0_Z}} \\ + & \mathbf{1_Z} \end{pmatrix}_{01} \begin{pmatrix} \mathbf{0_X} & + \\ \mathbf{1_X} & + \end{pmatrix}_{10} \begin{pmatrix} \mathbf{0_X} & + \\ + & \mathbf{1_Z} \end{pmatrix}_{01} \begin{pmatrix} + & \overline{\mathbf{0_Z}} \\ \mathbf{1_X} & + \end{pmatrix}_{10}$$

$$\text{CSS}: \quad \underline{0101} \quad 1010 \quad 1001 \quad \underline{0110} \qquad\qquad\qquad (f_{10})$$

7.　The error $\left(\overline{\mathbf{1_X}}, \mathbf{1_Z}\right)$ and also the error $\left(\mathbf{1_X}, \overline{\mathbf{1_Z}}\right)$ are detected with an error-free NO-QP $(\mathbf{0_X}, \mathbf{0_Z})$, as the first row in $f_8$ because the CSS that is produced by the error is none of the possible error-free CSS.

$$\begin{pmatrix} \mathbf{0_X} & \mathbf{0_Z} \\ \mathbf{1_X} & \mathbf{1_Z} \end{pmatrix} : \begin{pmatrix} + & \mathbf{0_Z} \\ + & \mathbf{1_Z} \end{pmatrix}_{01} \begin{pmatrix} \mathbf{0_X} & + \\ \mathbf{1_X} & + \end{pmatrix}_{10} \begin{pmatrix} \mathbf{0_X} & + \\ + & \mathbf{1_Z} \end{pmatrix}_{01} \begin{pmatrix} + & \mathbf{0_Z} \\ \mathbf{1_X} & + \end{pmatrix}_{10}$$

$$\begin{pmatrix} \mathbf{1_X} & \mathbf{1_Z} \\ \mathbf{0_X} & \mathbf{0_Z} \end{pmatrix} : \begin{pmatrix} + & \mathbf{1_Z} \\ + & \mathbf{0_Z} \end{pmatrix}_{01} \begin{pmatrix} \mathbf{1_X} & + \\ \mathbf{0_X} & + \end{pmatrix}_{10} \begin{pmatrix} \mathbf{1_X} & + \\ + & \mathbf{0_Z} \end{pmatrix}_{10} \begin{pmatrix} + & \mathbf{1_Z} \\ \mathbf{0_X} & + \end{pmatrix}_{01}$$

$$\text{CSS}: \quad 0101 \quad 1010 \quad 0110 \quad 1001$$

$$\begin{pmatrix} \mathbf{0_X} & \mathbf{0_Z} \\ \overline{\mathbf{1_X}} & \mathbf{1_Z} \end{pmatrix} : \begin{pmatrix} + & \mathbf{0_Z} \\ + & \mathbf{1_Z} \end{pmatrix}_{01} \begin{pmatrix} \mathbf{0_X} & + \\ \overline{\mathbf{0_X}} & + \end{pmatrix}_{00} \begin{pmatrix} \mathbf{0_X} & + \\ + & \mathbf{1_Z} \end{pmatrix}_{01} \begin{pmatrix} + & \mathbf{0_Z} \\ \overline{\mathbf{0_X}} & + \end{pmatrix}_{00}$$

$$\begin{pmatrix} \mathbf{1_X} & \mathbf{1_Z} \\ \overline{\mathbf{0_X}} & \mathbf{0_Z} \end{pmatrix} : \begin{pmatrix} + & \mathbf{1_Z} \\ + & \mathbf{0_Z} \end{pmatrix}_{01} \begin{pmatrix} \mathbf{1_X} & + \\ \overline{\mathbf{1_X}} & + \end{pmatrix}_{00} \begin{pmatrix} \mathbf{1_X} & + \\ + & \mathbf{0_Z} \end{pmatrix}_{10} \begin{pmatrix} + & \mathbf{1_Z} \\ \overline{\mathbf{1_X}} & + \end{pmatrix}_{11}$$

$$\text{CSS}: \quad 0101 \quad \underline{0000} \quad 0110 \quad \underline{0011} \qquad\qquad\qquad (f_8)$$

8.　The error $\left(\overline{\mathbf{1_X}}, \mathbf{1_Z}\right)$ and also the error $\left(\mathbf{1_X}, \overline{\mathbf{1_Z}}\right)$ are detected with an error-free NO-QP $(\mathbf{0_X}, \mathbf{0_Z})$, as the second row in $f_{12}$ because the CSS that is produced by the error is none of the possible error-free CSS.

$$\begin{pmatrix} \mathbf{1_X} & \mathbf{1_Z} \\ \mathbf{0_X} & \mathbf{0_Z} \end{pmatrix} : \begin{pmatrix} + & \mathbf{1_Z} \\ + & \mathbf{0_Z} \end{pmatrix}_{01} \begin{pmatrix} \mathbf{1_X} & + \\ \mathbf{0_X} & + \end{pmatrix}_{10} \begin{pmatrix} \mathbf{1_X} & + \\ + & \mathbf{0_Z} \end{pmatrix}_{10} \begin{pmatrix} + & \mathbf{1_Z} \\ \mathbf{0_X} & + \end{pmatrix}_{01}$$

$$\begin{pmatrix} \mathbf{0_X} & \mathbf{0_Z} \\ \mathbf{1_X} & \mathbf{1_Z} \end{pmatrix} : \begin{pmatrix} + & \mathbf{0_Z} \\ + & \mathbf{1_Z} \end{pmatrix}_{01} \begin{pmatrix} \mathbf{0_X} & + \\ \mathbf{1_X} & + \end{pmatrix}_{10} \begin{pmatrix} \mathbf{0_X} & + \\ + & \mathbf{1_Z} \end{pmatrix}_{01} \begin{pmatrix} + & \mathbf{0_Z} \\ \mathbf{1_X} & + \end{pmatrix}_{10}$$

$$\text{CSS}: \quad 0101 \quad 1010 \quad 1001 \quad 0110$$

$$\begin{pmatrix} \boxed{\mathbf{1_X}} & \mathbf{1_Z} \\ \mathbf{0_X} & \mathbf{0_Z} \end{pmatrix} : \begin{pmatrix} + & \mathbf{1_Z} \\ + & \mathbf{0_Z} \end{pmatrix}_{01} \begin{pmatrix} \boxed{\mathbf{0_X}} & + \\ \mathbf{0_X} & + \end{pmatrix}_{00} \begin{pmatrix} \boxed{\mathbf{0_X}} & + \\ + & \mathbf{0_Z} \end{pmatrix}_{00} \begin{pmatrix} + & \mathbf{1_Z} \\ \mathbf{0_X} & + \end{pmatrix}_{01}$$

$$\begin{pmatrix} \boxed{\mathbf{0_X}} & \mathbf{0_Z} \\ \mathbf{1_X} & \mathbf{1_Z} \end{pmatrix} : \begin{pmatrix} + & \mathbf{0_Z} \\ + & \mathbf{1_Z} \end{pmatrix}_{01} \begin{pmatrix} \boxed{\mathbf{1_X}} & + \\ \mathbf{1_X} & + \end{pmatrix}_{00} \begin{pmatrix} \boxed{\mathbf{1_X}} & + \\ + & \mathbf{1_Z} \end{pmatrix}_{11} \begin{pmatrix} + & \mathbf{0_Z} \\ \mathbf{1_X} & + \end{pmatrix}_{10}$$

$$\text{CSS}: \quad 0101 \quad \underline{0000} \quad \underline{0011} \quad 0110 \qquad\qquad (f_{12})$$

### 2.2. Error Correction Pre-Processing

In the error-detection cases exhibited above we have assumed that the non-orthogonal quantum pairs (NO-QP) $(\mathbf{0_X}, \mathbf{0_Z})$ and $(\mathbf{1_X}, \mathbf{1_Z})$ are error-free, which implies being able to detect any error in these NO-QP. In this section, we explain how to detect such errors using null-frames $f_7$ and unitary-frames $f_{11}$. To achieve this we must note that in the absence of error, $f_7$ produce CSS that only contain zeros while $f_{11}$ produces 0000 and 1100 under MRT.

#### 2.2.1. Null-Frame Errors

The error can arrive in several ways and we need a method to detect them, so the following cases can occur:

— Single error:

$$\begin{pmatrix} \boxed{\mathbf{0_X}} & \mathbf{0_Z} \\ \mathbf{0_X} & \mathbf{0_Z} \end{pmatrix}, \begin{pmatrix} \mathbf{0_X} & \mathbf{0_Z} \\ \boxed{\mathbf{0_X}} & \mathbf{0_Z} \end{pmatrix}, \begin{pmatrix} \mathbf{0_X} & \boxed{\mathbf{0_Z}} \\ \mathbf{0_X} & \mathbf{0_Z} \end{pmatrix}, \begin{pmatrix} \mathbf{0_X} & \mathbf{0_Z} \\ \mathbf{0_X} & \boxed{\mathbf{0_Z}} \end{pmatrix}$$

— Non-orthogonal error, two errors in different basis:

$$\begin{pmatrix} \boxed{\mathbf{0_X}} & \mathbf{0_Z} \\ \mathbf{0_X} & \boxed{\mathbf{0_Z}} \end{pmatrix}, \begin{pmatrix} \mathbf{0_X} & \boxed{\mathbf{0_Z}} \\ \boxed{\mathbf{0_X}} & \mathbf{0_Z} \end{pmatrix}$$

— Parallel error, two errors in the same basis:

$$\begin{pmatrix} \boxed{\mathbf{0_X}} & \mathbf{0_Z} \\ \boxed{\mathbf{0_X}} & \mathbf{0_Z} \end{pmatrix}, \begin{pmatrix} \mathbf{0_X} & \boxed{\mathbf{0_Z}} \\ \mathbf{0_X} & \boxed{\mathbf{0_Z}} \end{pmatrix}$$

As we will see right away, single and parallel errors will be detected as if they were non-orthogonal error using the algorithm for Detection of Parallel-Pair Errors (DPPE) that we explain next:

— Alice separates CSSL from null frames into the error-detected-null-frames and the error-free-null-frames just checking that CSS $\neq$ 0000. The last list contain, however, frames with hidden (parallel) errors. For example, consider Alice's null frame $\begin{pmatrix} \mathbf{0_X} & \mathbf{0_Z} \\ \mathbf{0_X} & \mathbf{0_Z} \end{pmatrix}$. If Bob's frame contains two errors, say $\begin{pmatrix} \mathbf{1_X} & + \\ \mathbf{1_X} & + \end{pmatrix}$ then the errors kept hidden since $(1100, x_1, x_2)$. Alice takes the row-indices $x_1$ and $x_2$ and she looks for them into the error-detected list (see Table 3).

— Two cases in the error-detected list reveals the errors in $x_1$ and $x_2$. The frame $\begin{pmatrix} \mathbf{1_X} & + \\ \mathbf{0_X} & + \end{pmatrix}$ with $(1010, x_1, *)$ where the first row is $x_1$, reveals an error in the frame but the result it is ambiguous because the frame $\begin{pmatrix} \mathbf{0_X} & + \\ \mathbf{1_X} & + \end{pmatrix}$ also produces CSS = 1010. Thus, the result is inconclusive because the CSS does not indicate if the error is in the first or the second row. However, Alice keeps searching into the list of errors and she finds $(1100, x_1, *)$ which comes from the frame $\begin{pmatrix} \mathbf{1_X} & + \\ + & \mathbf{1_X} \end{pmatrix}$ where the first row contains $x_1$ and the second row also contains an error. Interestingly, this CSS reveals the presence of a non-orthogonal error. Similarly, Alice finds another label that exhibits $x_2$ and the parallel error is detected using a non-orthogonal error which applies for single errors too.

The following aspects must be remarked here:

1. Given an error rate in the quantum channel, it is to be expected that about half of the errors will occur in the first quantum state of $\left(\overline{\mathbf{0_X}}, \mathbf{0_Z}\right)$ and the other half in the second state $\left(\mathbf{0_X}, \overline{\mathbf{0_Z}}\right)$. Therefore, the method described to detect single and parallel errors in null-frames is completely feasible.

2. The algorithm detailed above allows finding all the errors in the null frames, but it does not tell us which of the two non-orthogonal states is the error. To find the position of the error, Alice must use an error free NO-QP say $y_1 = (\mathbf{0_X}, \mathbf{0_Z})$ from the list of error-free-null-frames. Then she finds $(0101, x_1, y_1)$ which reveals the error is in the first state while $(1010, x_1, y_1)$ unveils the error in the second state (see Table 3).

How should it be clear, in all cases, Alice must be able to identify the position of the error to perform reconciliation successfully. Importantly here is that detecting the position of the error also allows Alice to find MR when using this error-row inside a frame. For this purpose and assuming Alice has detected all the errors, consider the following frame cases:

1. First and second rows without errors. Alice applies the usual frame-based sifting algorithm identifying MR in each case.

2. First and second rows with errors. Since error-detection reveals the position of the error, Alice identifies MR straightforward.

3. Error-free (first/second) row and error-detected (second/first) row. In the next lines we discuss this case.

Suppose Alice has detected all the errors and she must guess MR when she sends $\begin{pmatrix} \mathbf{0_X} & \mathbf{1_Z} \\ \mathbf{0_X} & \mathbf{0_Z} \end{pmatrix}$ and Bob gets $\begin{pmatrix} + & \mathbf{1_Z} \\ \overline{\mathbf{1_X}} & + \end{pmatrix}$ thus he returns to Alice CSS = 1100. But Alice knows the following facts:

— CSS comes from $f_9$.

— The first row is error-free but the first state of the second row is error-detected, that is $\left(\overline{\mathbf{0_X}}, \mathbf{0_Z}\right)$.

Then Alice tests $f_9$ under MRT (see Table 1) given $\left(\overline{\mathbf{0_X}}, \mathbf{0_Z}\right)$ and CSS = 1100, thus she concludes that the unique MR that matches CSS is under MR = 11.

**Table 3.** Analysis of $f_7$ frame set. Separate lists of error-detected-null-frames and error-free-null-frames. The symbol $*$ represents an arbitrary NO-QP index.

| Error-Detected | | | | Error-Free | | | |
|---|---|---|---|---|---|---|---|
| **CSS ! 0** | $i_1$ | $i_2$ | **Comment** | **CSS = 0** | $i_1$ | $i_2$ | **Comment** |
| 1010 | $*$ | $*$ | | 0000 | $x_1$ | $x_2$ | hidden error |
| 1100 | $x_1$ | $*$ | double-error-detection | 0000 | $y_1$ | $y_2$ | |
| 1100 | $x_2$ | $*$ | double-error-detection | 0000 | $*$ | $*$ | |
| 1010 | $*$ | $*$ | | 0000 | $*$ | $*$ | |
| 1001 | $*$ | $*$ | | 0000 | $*$ | $*$ | |
| 0101 | $x_1$ | $y_1$ | first-state-error | 0000 | $*$ | $*$ | |
| 1010 | $x_2$ | $y_1$ | second-state-error | 0000 | $*$ | $*$ | |

### 2.2.2. Unitary-Frame Errors

Frames $f_{11}$ behave similarly as frames $f_7$, so we present the summary results in Table 4. As far as we go, we are able to detect $\left(\overline{0_X}, 0_Z\right)$, $\left(0_X, \overline{0_Z}\right)$, $\left(\overline{1_X}, 1_Z\right)$, $\left(1_X, \overline{1_Z}\right)$ errors.

**Table 4.** Analysis of $f_{11}$ frame set. Separate lists of error-detected-unitary-frames and error-free-unitary-frames. The symbol $*$ represents an arbitrary NO-QP index.

| Error-Detected | | | | Error-Free | | | |
|---|---|---|---|---|---|---|---|
| **CSS** | $i_1$ | $i_2$ | **Comment** | **CSS** | $i_1$ | $i_2$ | **Comment** |
| 1010 | $*$ | $*$ | | 0000 | $x_1$ | $x_2$ | hidden error |
| 0011 | $x_1$ | $*$ | double-error-detection | 1100 | $y_1$ | $y_2$ | |
| 0011 | $x_2$ | $*$ | double-error-detection | 1100 | $*$ | $*$ | |
| 0101 | $*$ | $*$ | | 1100 | $*$ | $*$ | |
| 1001 | $*$ | $*$ | | 0000 | $*$ | $*$ | |
| 1010 | $x_1$ | $y_1$ | first-state-error | 0000 | $*$ | $*$ | |
| 1001 | $x_2$ | $y_1$ | second-state-error | 1100 | $*$ | $*$ | |

### 2.3. Reconciliation Algorithm

To close this section let us summarize the steps of the reconciliation algorithm. Table 5 shows the error detection results using regular frames.

1.  Identify $(0_X, 0_Z)$ and $\left(\overline{0_X}, 0_Z\right)$, $\left(0_X, \overline{0_Z}\right)$ errors in the set of $f_7$. Identify single and parallel errors using DPPE algorithm.
2.  Identify $(1_X, 1_Z)$ and $\left(\overline{1_X}, 1_Z\right)$, $\left(1_X, \overline{1_Z}\right)$ errors in the set of $f_{11}$. Identify single and parallel errors using DPPE algorithm.
3.  Identify MR using $(0_X, 0_Z)$ $\left(\overline{0_X}, 0_Z\right)$, $\left(0_X, \overline{0_Z}\right)$ and $(1_X, 1_Z)$ $\left(\overline{1_X}, 1_Z\right)$, $\left(1_X, \overline{1_Z}\right)$ in $f_8$, $f_{12}$.
4.  Identify $(0_X, 1_Z)$, $(1_X, 0_Z)$ and $\left(\overline{0_X}, 1_Z\right)$, $\left(1_X, \overline{0_Z}\right)$ errors in $f_9$, $f_{10}$, $f_{13}$, $f_{14}$ using $(0_X, 0_Z)$, $\left(\overline{0_X}, 0_Z\right)$, $\left(0_X, \overline{0_Z}\right)$. Identify MR in $f_9$, $f_{10}$, $f_{13}$, $f_{14}$.
5.  Identify $(0_X, 1_Z)$, $(1_X, 0_Z)$ and $\left(0_X, \overline{1_Z}\right)$, $\left(\overline{1_X}, 0_Z\right)$ errors in $f_2$, $f_6$, $f_3$, $f_4$ using $(1_X, 1_Z)$, $\left(\overline{1_X}, 1_Z\right)$, $\left(1_X, \overline{1_Z}\right)$, $(0_X, 1_Z)$, $(1_X, 0_Z)$, $\left(\overline{0_X}, 1_Z\right)$, $\left(1_X, \overline{0_Z}\right)$. Identify MR in $f_2$, $f_6$, $f_3$, $f_4$.
6.  Identify MR in $f_1$, $f_5$ using $(0_X, 1_Z)$, $(1_X, 0_Z)$, $\left(0_X, \overline{1_Z}\right)$, $\left(\overline{0_X}, 1_Z\right)$, $\left(1_X, \overline{0_Z}\right)$, $\left(\overline{1_X}, 0_Z\right)$.

**Table 5.** Error detection using regular frames.

| CSS | Frame | MR | Error Detection |
|:---:|:---:|:---:|:---:|
| 1010 | $f_2, f_6$ | 00 | $\left(\overline{1_X}, 0_Z\right)$ |
| 0110 | $f_2$ $f_6$ | 10 11 | $\left(\overline{1_X}, 0_Z\right)$ |
| 1001 | $f_3$ $f_4$ | 11 10 | $\left(0_X, \overline{1_Z}\right)$ |
| 0101 | $f_3, f_4$ | 01 | $\left(0_X, \overline{1_Z}\right)$ |
| 1010 1001 | $f_9$ | 00 10 | $\left(\overline{0_X}, 1_Z\right)$ |
| 1010 1001 | $f_{13}$ | 00 11 | $\left(\overline{0_X}, 1_Z\right)$ |
| 0101 1001 | $f_{10}$ | 01 10 | $\left(1_X, \overline{0_Z}\right)$ |
| 0101 0110 | $f_{14}$ | 01 10 | $\left(1_X, \overline{0_Z}\right)$ |

As has been demonstrated so far, errors can be detected regardless of the number of errors. Thus, the gain of the secret bits does not depend on the error rate of the quantum channel.

## 3. The Throughput of Frame Reconciliation

Not all the frames are converted into secret bits. In [21] we derived the throughput as $\frac{1}{4}\left(\frac{1}{2}(1-e) + \frac{1}{6}e\right)$ that reaches a maximum gain of $\frac{1}{8}\binom{n}{2}$ when $e = 0$. Taking into account Table 2, we arrive to the throughput Equation (1) of frame reconciliation $T$ where $n$ is the number or Double Matching Detection Events.

$$
\begin{aligned}
T &= \frac{1}{16}\left(4 \cdot \frac{1}{2} + 8 \cdot \frac{3}{4}\right)\binom{n}{2} \\
&= \frac{1}{2}\binom{n}{2} = \frac{1}{2}\frac{n(n-1)}{2} \\
&\sim \frac{1}{4}n^2
\end{aligned}
\tag{1}
$$

Computing the photonic gain of double detection events at Bob's side as $Q_{(+,+)} = (1 - e^{-\mu})^2$ (neglecting the losses generated by the quantum channel and the losses of the optical detection system), we derived Equation (2) where $N$ is the total number of quantum pulses sent by Alice to Bob. As a result, the number of secret bits grows doubly quadratically as a function of the number of quantum pulses $N$.

$$
T = \frac{1}{4}(1 - e^{-\mu})^4 N^4
\tag{2}
$$

One of the biggest challenges posed by the Photon Number Splitting (PNS) attack is that the number of photons per pulse ($\mu$) should not be increased because an attacker can split the pulse and store a copy of it. However, in frame-based reconciliation, the secret bits do not result only from the quantum pulses that arrive to Bob's detector, but from the

double detection events that occur at Bob's station. So the security of our approach does no depend on the photon mean $\mu$ of the quantum pulse neither the channel error-rate $e$.

## 4. Immunity to Quantum Attacks

Produce a double detection event does not depend on the transmittance of the quantum channel but in the quantum probability. This property gives immunity to the quantum key distribution protocol when it relies on double detection event as the vehicle to transmit a secret bit. Since the reconciliation efficiency does not depend on the quantum channel error rate, we will start this section by looking at the effect of quantum channel noise on error production. Later we explain why the frame-based QKD is immune to the Intercept-Resend attack (IR), the Photon Number Splitting attack (PNS) and the Quantum Bases Choice attack (BC). Our research work is still in progress to extend the discussion and demonstrate security of our method over other more general quantum attacks as collective attacks.

### 4.1. Errors in the Quantum Channel

Whatever the type of the noise in the quantum channel, be it rotation noise or dephasing noise [22], it can be interpreted as a Bloch sphere rotation axis causing a variation in the polarization of the quantum state [23], thus producing errors in the information encoded in it. Suppose, for example, that due to the effect of channel noise, state $|0_Z\rangle$ moves to $|1_X\rangle$ and state $|0_X\rangle$ becomes $|1_Z\rangle$ as represented in Figure 4. If Bob uses the basis $X$ to measure the pair of non-orthogonal states $(|0_X\rangle, |0_Z\rangle)$, there are two probable outcomes:

— $|1_X\rangle$ as a result of a double matching detection event which according to the reconciliation method is taken as an error.
— $(|0_X\rangle, |1_X\rangle)$ caused by a double non-matching detection event that is useless to perform reconciliation.

The same result applies for the $Z$ basis (see Figure 4). The most important conclusion from this analysis is that $|0_X\rangle$ can never be accepted as $|1_X\rangle$ since it is interpreted as an error. Such behavior comes from the fact that a pair of orthogonal states hold a single bit of information. Let us now proceed to the analysis of the behavior of the protocol in the presence of attacks on the QKD system.

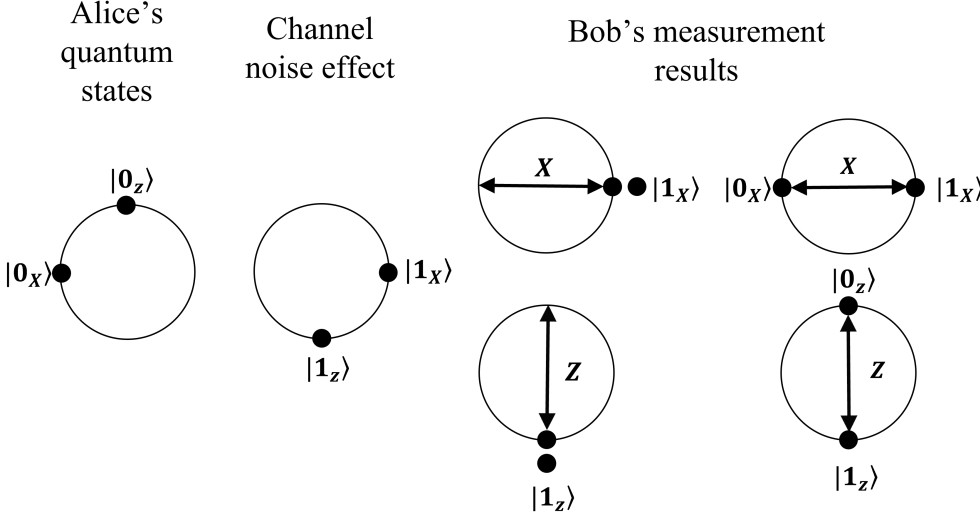

**Figure 4.** We see in the (2-dimensional) Bloch sphere the effect generated by the noise of the quantum channel on the pair of non-orthogonal quantum states.

### 4.2. The Intercept and Resend Attack (IR)

Eve must measure each pair of non-orthogonal quantum pulses that crosses the quantum channel and produce a double matching detection event, then according to the result obtained from her measurement, Eve sends another pair of non-orthogonal quantum

pulses to Bob. In addition, Eve must ensure that both states that she forwards to Bob's station are not lost on the quantum channel, but assuming she can overcome this difficulty, Eve's final gain, as indicated in Table 6 is 0.25.

**Table 6.** Eve is forced to produce a double detection event, then she must guess Bob's basis which occurs with 0.5 probability, so Eve's final probability is 0.25.

| Alice | Eve | Bob |
|:---:|:---:|:---:|
| $Q_{(+,+)}$ | 0.5 DMDE | 0.5 **X** basis |
| | | 0.5 **Z** basis |

### 4.2.1. The Photon Number Splitting Attack (PNS)

Eve obtains a copy of the quantum states that Bob receives in his optical station and stores them in a quantum memory. However, the probability that Eve gets a double matching detection event as registered by Bob is 0.5. In addition, Eve must measure by choosing between two different measurement bases (**X** or **Z**), so the final gain of the attack is 0.25 (see Table 7).

**Table 7.** Eve is required to produce a double matching detection event as Bob. In addition, Eve must choose the appropriate basis which occurs with 0.5 probability, so Eve's final probability is 0.25.

| Alice | Bob | Eve |
|:---:|:---:|:---:|
| $Q_{(+,+)}$ | 0.5 DMDE | 0.5 **X** basis |
| | | 0.5 **Z** basis |

### 4.2.2. The Quantum Measurement Bases Choice Attack (BC)

The attacker could use other quantum measurement bases to gain more information, for example Eve could use the pair of measurement bases **X** + **Z**, **X** − **Z**. Now, suppose Bob has registered a double matching detection event and Eve has a copy of those states then she could get Bob's information with a probability of 0.28. This is so because Eve chooses one of the measurement bases **X** + **Z** or **X** − **Z** with a probability of 0.5. But the non-matching detection events are ambiguous for Eve, which occurs with a probability of 0.37. Rather, she gets a double matching event with probability 0.56. As a result, the probability of getting Bob's information is 0.28 (see Table 8).

**Table 8.** Eve is required to produce a double matching detection event as Bob with probability 0.56. In addition, Eve must choose the appropriate basis which occurs with probability of 0.5, so Eve's final probability is 0.28.

| Alice | Bob | Eve |
|:---:|:---:|:---:|
| $Q_{(+,+)}$ | 0.56 DMDE | 0.5 **X** basis |
| | | 0.5 **Z** basis |

We must highlight that one the main advantages of the immunity to described quantum attacks, is that the mean photon value $\mu$ in Equation (2) can be properly increased in the quantum regime, so that longer distance can be achieved in QKD link. Further on, the number of secret bits grows doubly quadratic in the number of quantum pulses.

## 5. Discussion

Table 9 shows according to [24] a general comparison between some of the most representative reconciliation algorithms where we have included the introduced method. Although we are currently working in the software implementation of the algorithm, we

base our evaluation in the following criteria: the algorithm requires only four message exchanges as can be seen in Figure 3. The effectiveness of the protocol is perfect since all errors are corrected regardless of the quantum channel error rate.

The QKD protocol runs in a single process that includes quantum communication and classical post-processing based on two-order frames. The throughput grows doubly quadratically as a function of the number of quantum pulses and the runtime requires only milliseconds as demonstrated in our previous software implementation. Furthermore, our method does not require redundant bits to be sent, and there is no need to estimate quantum channel error rate while is invariant against burst errors. We present an appendix that describes the execution of the QKD protocol and the frame-based reconciliation algorithm (see Appendix A).

**Table 9.** Comparison of main reconciliation algorithms based on [24]. Complexity refers to computational or communication complexity. Effectiveness is the corrective percentage efficiency of the protocol. Throughput is the secret key rate measured in bits per second. Runtime is the number of seconds required to execute the algorithm.

| Algorithm | Complexity | Effectiveness | Throughput | Runtime |
|---|---|---|---|---|
| Cascade | High | Medium | Low | Medium |
| Winnow | Medium | Low | High | Low |
| LDPC | High | High | Medium | High |
| This work | Low | High | High | Low |

## 6. Conclusions

We introduced a method to achieve complete reconciliation in Quantum Key Distribution which identifies the transmitted errors in a reverse reconciliation that corrects 100% of the errors that is invariant with respect to the error rate of the quantum channel.

The QKD protocol, which is based on sending pairs of non-orthogonal quantum states and reconciliation through frames of two-order, is executed in a single process that includes quantum communication and classical post-processing. Furthermore, it does not require the sending of redundant bits, nor is it necessary to estimate the error rate of the quantum channel.

At least theoretically, the number of secret bits grows doubly quadratically as a function of the number of quantum pulses sent by Alice because the mean photon value can be properly increased in the quantum regime, so that longer distance can be achieved in QKD system.

**Author Contributions:** L.A.L.-P. conceived of the presented idea, he developed the theoretical formalism, J.M.L.-R. supervised the project and contributed to the interpretation of the results. All authors have read and agreed to the published version of the manuscript.

**Funding:** This research was funded by National Council of Science and Technology of Mexico (CONACyT) and Center for Research and Advanced Studies of the National Polytechnic Institute of Mexico (Cinvestav-IPN).

**Data Availability Statement:** The data presented in this study are available within the article.

**Conflicts of Interest:** The authors declare no conflict of interest in this article.

## Appendix A

This appendix contains a simple execution of the protocol using the CSS-based reconciliation algorithm. To simplify the discussion in Figure A1, we assume that the channel causes errors but does not lose any quantum pulses, in addition the pairs of non-orthogonal states are sent sequentially and all of them produce double matching detection events. Let us describe the steps of the QKD protocol:

1. Alice creates NO-QPL (the Non-Orthogonal Quantum Pair List) and sends, one by one, each NO-QP (the Non-Orthogonal Quantum Pair) across QC (the Quantum Channel).

In the simulation NO-QP are $1:(0_X, 0_Z)$, $2:(0_X, 0_Z)$, $3:(1_X, 1_Z)$, $4:(1_X, 1_Z)$, $5:(0_X, 1_Z)$, $6:(0_X, 1_Z)$.

2. Bob chooses randomly the measurement basis: **X** or **Z**, that he will use to measure both states inside NO-QP. After Bob registers DDE (the Double Detection Events) he sends DDEL (the Double Detection Event List) to Alice. In this example, we assume that all NO-QP produce DDE, so DDEL = 1 . . . 6.

3. Alice receive DDEL from QC, she creates FAIL (the Frame Arrangement Information List) and sends it to Bob. Here we have DDEL = 1:(1, 2), 2:(3, 4), 3:(1, 3), 4:(1, 4), 5:(2, 3), 6:(2, 4),7:(3, 1), 8:(4, 1), 9:(3, 2), 10:(4, 2), 11:(5, 1), 12:(5, 2),13:(1, 5), 14:(2, 5), 15:(6, 3), 16:(6, 4), 17:(3, 6), 18:(4, 6).

4. Bob receives FAIL and he computes CSSL (the Composed Sifting String List). Then he returns CSSL to Alice that in our case corresponds to CSSL= 1:1100, 2:1100, 3:1100, 4:0000, 5:0000, 6:1100, 7:1100, 8:0000, 9:0000, 10:1100, 11:0101, 12:1001, 13:0101, 14:1001, 15:1001, 16:0101, 17:1001, 18:0101.

5. Alice detect errors and identifies MR in regular frames. Alice sends FDL (the Frame to Delete List) to Bob. Alice achieves error correction as indicated by the reconciliation algorithm discussed in Section 2.3 and illustrated in Table A1. It should be noted that for the simulation we assume a conservative channel so FDL is empty.

6. Bob eliminates the frames indicated in FDL, then he creates SeS using MRT as written in Table 1. In the example we derive 16 secret bits.

As shown in Table A1, 16 of 18 frames are in error, so $e = 0.88$. However, 16 bits are distilled while only 6 double detection events have been registered at Bob's station, demonstrating the high throughput of secret bits that the protocol produces.

A relevant case to be taken into account is when the frame contains double errors. We must clarify that Alice knows in advance that the frame contains two errors, so to get the MR, Alice only has to apply Table 1. This is because the MRs of the error-free frame do not contradict the MRs of the two-error frame. On the contrary, if the frame contains only one error it is necessary that given the error, Alice determines which is the MR that satisfies the CSS.

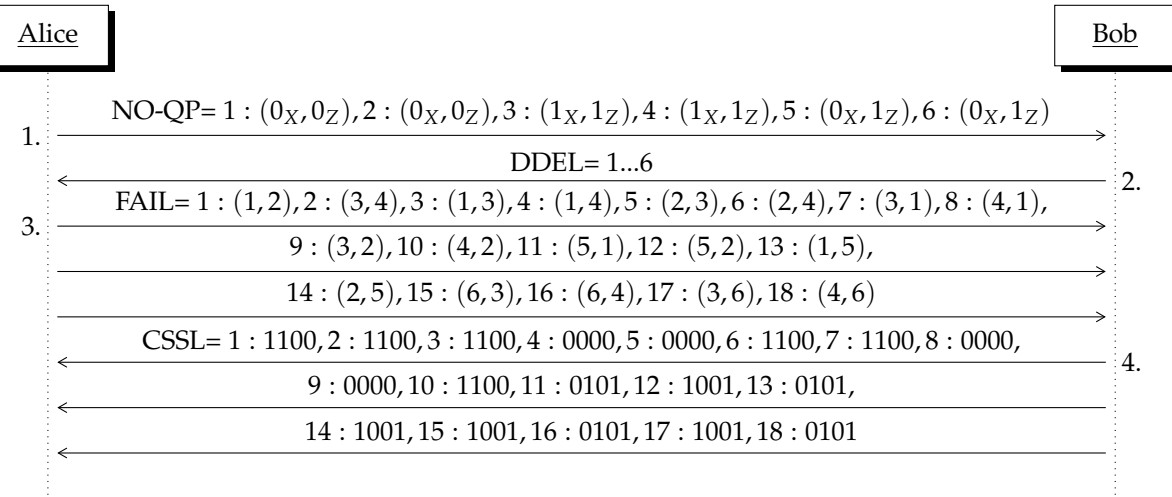

**Figure A1.** An illustrative execution of the protocol. In this simulation 16 of 18 frames are in error.

**Table A1.** Alice's reconciliation process. Error detection is indicated with an overbracket symbol.

| Item | Frame | NO-QP Index | CSS | MR | Error-Detection |
|------|-------|-------------|-----|-----|-----------------|
| 1. | $f_7$ | 1, 2 | 1100 | - | ⎴1, ⎴2 |
| 2. | $f_{11}$ | 3, 4 | 1100 | - | 3 , 4 |
| 3. | | 1, 3 | 1100 | 11 | ⎴1, 3 |
| 4. | | 1, 4 | 0000 | 01 | ⎴1, 4 |
| 5. | $f_8$ | 2, 3 | 0000 | 00 | ⎴2, 3 |
| 6. | | 2, 4 | 1100 | 10 | ⎴2, 4 |
| 7. | | 3, 1 | 1100 | 10 | 3, ⎴1 |
| 8. | | 4, 1 | 0000 | 01 | 4, ⎴1 |
| 9. | $f_{12}$ | 3, 2 | 0000 | 00 | 3, ⎴2 |
| 10. | | 4, 2 | 1100 | 11 | 4, ⎴2 |
| 11. | | 5, 1 | 0101 | 01 | ⎴5, ⎴1 |
| 12. | $f_9$ | 5, 2 | 1001 | 11 | ⎴5, ⎴2 |
| 13. | | 1, 5 | 0101 | 01 | ⎴1, ⎴5 |
| 14. | $f_{13}$ | 2, 5 | 1001 | 10 | ⎴2, ⎴5 |
| 15. | | 6, 3 | 1001 | 11 | ⎴6, 3 |
| 16. | $f_3$ | 6, 4 | 0101 | 01 | ⎴6, 4 |
| 17. | | 3, 6 | 1001 | 11 | 3, ⎴6 |
| 18. | $f_4$ | 4, 6 | 0101 | 01 | 4, ⎴6 |

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
