# Peer review of "Perfect Reconciliation in Quantum Key Distribution with Order-Two Frames"

_symmetry, doi:10.3390/sym13091672_

Round 1

Reviewer 1 Report

The authors have developed an error reconciliation method for Quantum Key Distribution

in a noisy quantum channel. They state that this technique increases quadratically the secret key rate as a function of the double matching detection events and doubly quadratically in the number of the quantum pulses.

The subject is interesting however the paper needs to be revised before any decision.

My comments are as following:

  • The quantum cryptographic key distribution should be explained in more detail it could be in an appendix
  • The authors should include more details on the discussion of section 4 qualitative study and simulations
  • The fidelity should be calculated in a noisy environment.

Author Response

The authors deeply appreciate the suggestion of Reviewer 1. Please find in the attached file our reply to the points raised.

Reviewer 2 Report

This manuscript proposed an error reconciliation method for quantum key distribution (QKD) protocols. The authors claimed that their approach can correct 100% of errors generated in regular binary frame. However, there are some questions and suggestions required clarifications and improvements before this manuscript can be reconsider for publication in Symmetry.

  1. In the third paragraph of the introduction. The term “post-quantum” usually refers to classical cryptosystems that are thought to be secure against the attack by a quantum computer. The authors seem to reinvent its meaning to any QKD scheme.
  2. For the self-citations of reference [3] and [4], I do not see their relevance to the problem of this paper. Furthermore, it is 2021, the works published in more than five years ago should not be consider as “new” works, should it?
  3. In the introduction, it will be better to give a general definition of error reconciliation before directly addressed it to some related works.
  4. This work seems to be highly based upon the authors previous works in reference [22] and [23]. Can the authors address the differences between the core idea of this work versus those in their previous works?
  5. A lot of notations and abbreviations should be defined when they appeared the first time. For example, the minus “-” and plus “+” that first appeared in Fig. 1 and 2, respectively. On the other hand, for those abbreviations (DDEL, FAIL, CSSL, FDL, …) in Fig. 3, it will be better to put legends to provide their full names.
  6. The problem of error reconciliation is not new. Therefore, it will be better to provide a table with comparisons between the proposed method and some representative existing methods.
  7. To show the proposed work has some significant contributions to the community, it is better to add some performance related discussions such as why the proposed method can resist any type of errors in quantum channel? For example, can it resist the collective noises such as the dephasing noises and the rotation noises happened in quantum channels?
  8. The security analysis is too brief. There are more types of existing quantum attacks. Unless these two attacks are the ones that are most relevant to error reconciliation. Even so, it is inappropriate to just provide a self-citation to the authors' previous work [23] to simplify the discussions of other attacks.

Author Response

The authors express their gratitude to Reviewer 2 for his comments and suggestions. Please find in the attached file our specific answers. 

Round 2

Reviewer 1 Report

The revised version  is ok, I accept it

Reviewer 2 Report

The authors have revised their manuscript according to the reviewers' comments. The revisions have made significant clarifications and improvements. Therefore, I would like recommend it for publication in Symmetry.